# Research on CAPTCHAs Targeted at AI: Human–Easy, Model–Hard Visual Tasks for the AI Era

## Abstract

As generative artificial intelligence advances rapidly, traditional CAPTCHAs designed to distinguish humans from machines are losing efficacy. State-of-the-art deep learning systems now solve conventional challenges such as distorted text recognition with near-perfect accuracy. Motivated by this, we explore "AI-targeted CAPTCHAs"—challenge tasks that humans pass easily but multimodal models find difficult. Building on a review of prior work, we posit two cognitive pathways that contemporary models rely on: a "linguistic path" versus a "perceptual path." Guided by these hypotheses, we design five simple, highly intuitive visual question–answer tasks to systematically compare humans with leading multimodal models. Each task pairs a single image with a single question and covers color discrimination, size comparison, combined distractors, and counting legs on birds or fingers on human hands. We evaluate five mainstream multimodal systems under the same conditions as human participants and test two main hypotheses plus one sub-hypothesis. Results show: (1) on single-feature tasks such as color discrimination, top models approach human-level performance; however, for size comparison and combined-difference tasks that require low-level visual perception, model accuracy collapses to near zero while humans perform almost perfectly; (2) in bird-leg and hand-finger counting tasks, models frequently default to stereotyped prior knowledge, achieving 20% accuracy or lower, whereas humans rely on the image and score near 100%; (3) models recognize abnormal human fingers slightly better than abnormal bird legs, supporting sub-hypothesis H1.1 that models handle common human limbs better than non-typical species. These differences confirm that current vision–language models primarily follow a linguistic path for image understanding and lack a human-like low-level perceptual path for processing obvious visual information. The findings quantify the limits of current multimodal systems and demonstrate the feasibility of constructing new CAPTCHAs from such "AI-hard" tasks.

**Keywords:** AI-targeted CAPTCHA; vision–language models (VLMs); visual perception; common-sense/confirmation bias; human–AI performance gap; multimodal evaluation.

## 1 Introduction

Artificial intelligence has made remarkable gains in visual recognition and multimodal understanding. Deep neural networks now rival or surpass humans on standard benchmarks [LeCun et al., 2015, Russakovsky et al., 2015]. At the same time, the long-standing need to distinguish humans from machines in security-critical settings has not diminished. Classic CAPTCHAs—particularly text-distortion variants—were originally predicated on tasks that were considered easy for humans but hard for machines. Yet reCAPTCHA-style schemes have steadily lost security margin as machine

vision improved [von Ahn et al., 2008]. This raises a central question for the multimodal era: Which challenge types can sustainably remain human-easy but model-hard?

We study that question by proposing an AI-targeted CAPTCHA design space grounded in a two-pathway account of image question answering. Contemporary vision–language systems often appear to follow a primarily *linguistic path*: rather than adducing visual evidence, they default to high-frequency language patterns and commonsense templates learned during pretraining—e.g., assuming that a bird has two legs or a hand has five fingers—even when the image shows otherwise [Nickerson, 1998, Geirhos et al., 2020, Schulze-Buschoff et al., 2024]. Humans, by contrast, typically rely first on a *perceptual path*: fast, low-level mechanisms encode basic features such as color, size, location, and numerosity in parallel before higher-level inference engages [Treisman and Gelade, 1980, Itti and Koch, 2001, Rosenholtz, 2014, Dehaene, 2011, Marr, 1982, Wolfe and Horowitz, 2017]. This asymmetry resonates with Moravec's paradox, whereby abstract reasoning can be comparatively easy for machines while low-level perception remains difficult [Moravec, 1988].

Guided by this framework, we articulate two main hypotheses and one sub-hypothesis. **H1 (linguistic-bias)**: When a task requires direct inspection of low-level visual evidence, current models will perform poorly because they over-weight linguistic priors. **H2 (perceptual-deficit)**: Multimodal systems lack a robust human-like perceptual pathway and will systematically err when image evidence conflicts with commonsense. **H1.1 (sub-hypothesis)**: On limb-counting tasks, models will do slightly better with human hands than with birds, reflecting training exposure and tuning to human imagery [Schulze-Buschoff et al., 2024].

## 2 Related Work

### 2.1 CAPTCHAs and the human–machine gap

Early CAPTCHAs exploited gaps between human and machine vision to authenticate users at scale. reCAPTCHA famously coupled security with human-based character recognition, leveraging human effort to transcribe ambiguous text [von Ahn et al., 2008]. The advent of modern deep learning, however, has eroded the hardness assumptions of text-based CAPTCHAs [LeCun et al., 2015, Russakovsky et al., 2015], motivating new challenge families that do not hinge on distorted text.

### 2.2 Low-level perception and attention in human vision

A large literature shows that humans can extract basic visual features (e.g., color, size, orientation, location, and numerosity) rapidly and in parallel, often with little or no attentional load [Treisman and Gelade, 1980, Itti and Koch, 2001, Wolfe and Horowitz, 2017, Rosenholtz, 2014, Dehaene, 2011, Marr, 1982]. Bottom-up salience and top-down task goals jointly guide selection [Desimone and Duncan, 1995], and even in clutter, peripheral summary statistics support efficient search for obvious outliers [Rosenholtz, 2014]. These properties predict that tasks grounded in low-level evidence will remain trivial for healthy adults.

### 2.3 Shortcut learning and linguistic bias in modern models

Despite strong benchmark performance, deep networks often exploit statistical shortcuts that correlate with labels without supporting genuine perception or reasoning [Geirhos et al., 2020]. In multimodal systems, the language component can dominate, yielding answers that align with high-frequency commonsense rather than the specific image—a hallmark of confirmation bias [Nickerson, 1998]. Recent evaluations document persistent gaps in intuitive visual cognition and perception across state-of-the-art models [Schulze-Buschoff et al., 2024].

### 2.4 Security applications leveraging human–AI asymmetries

A complementary literature in machine learning security studies adversarial failures of vision models [e.g., Akhtar and Mian, 2018, Goodfellow et al., 2015]. While many such perturbations are not user-friendly for authentication, the broader lesson is salient: today's systems exhibit systematic perceptual weaknesses. Our proposal diverges by enforcing a human-ease constraint and by favoring visible, low-level cues and light commonsense conflict as the lever for separating humans from machines.

## 2.5  Summary and positioning

The decline of text-based CAPTCHAs, the psychology of pre-attentive perception, and shortcut learning in modern networks converge on the same design principle: build challenges that require direct, low-level inspection of the image, particularly in scenarios that conflict mildly with commonsense. The present study operationalizes this principle through five minimal task families and human–model comparisons, providing an immediately usable recipe for AI-targeted CAPTCHAs.

## 3  Research Hypotheses

- **H1 (Linguistic-bias hypothesis).** Multimodal models primarily rely on linguistic patterns and memorized commonsense when answering visual questions; thus they will perform poorly on tasks that require direct, low-level perceptual processing, while potentially appearing competitive on tasks with clear semantic cues.

- **H2 (Perceptual-deficit hypothesis).** Current models lack a robust human-like perceptual pathway; when visual evidence conflicts with their prior commonsense, they display systematic biases, often producing answers consistent with memory rather than the image.

- **H1.1 (Sub-hypothesis).** On tasks invoking commonsense about familiar human anatomy versus non-human species, models will perform slightly better on human hands than on birds, reflecting greater exposure and potential tuning toward human imagery in training [Schulze-Buschoff et al., 2024].

## 4  Methods

### 4.1  Task Design

We created five visual question–answer task families, each presented as a single image paired with a single question. Participants answered based solely on the provided image. The task families are: (A) size difference (odd-one-out: one item slightly larger), (B) color difference (odd-one-out: one item subtly different color; RGB distance $\approx 45$), (C) combined distractors (one size-different item and one color-different item), (D) counterfactual animals (counting visible bird legs), and (E) counterfactual human hands (counting visible fingers).

### 4.2  Participants and Models

Five adult volunteers completed 50 items each (10 per task family) with no time limit and a single allowed response per item. We evaluated five multimodal models under identical conditions: OpenAI GPT-5, Anthropic Claude-Sonnet-4 (20250514), Google Gemini 2.5-Pro, ByteDance Doubao-seed-1.6 (250615), and Alibaba Qwen 2.5-VL-72B-Instruct. All prompts were in Chinese to avoid language confounds, and models returned constrained-format answers (e.g., "r3c4", or an integer count).

### 4.3  Stimuli

For A–C we generated $5 \times 5$ arrays with a program that guaranteed a single valid odd-one-out per image and auto-recorded keys. For D–E we curated AI-generated photos depicting controlled limb abnormalities while avoiding privacy concerns. All materials adhered to ethical guidelines.

### 4.4  Procedure

Humans solved all 50 items once each. For models, we ran up to 10 independent answer attempts per item to gauge response stability. We computed average accuracy per task family for humans and for each model, and we ran Fisher's exact tests and $\chi^2$ tests to assess significance of human–AI differences.

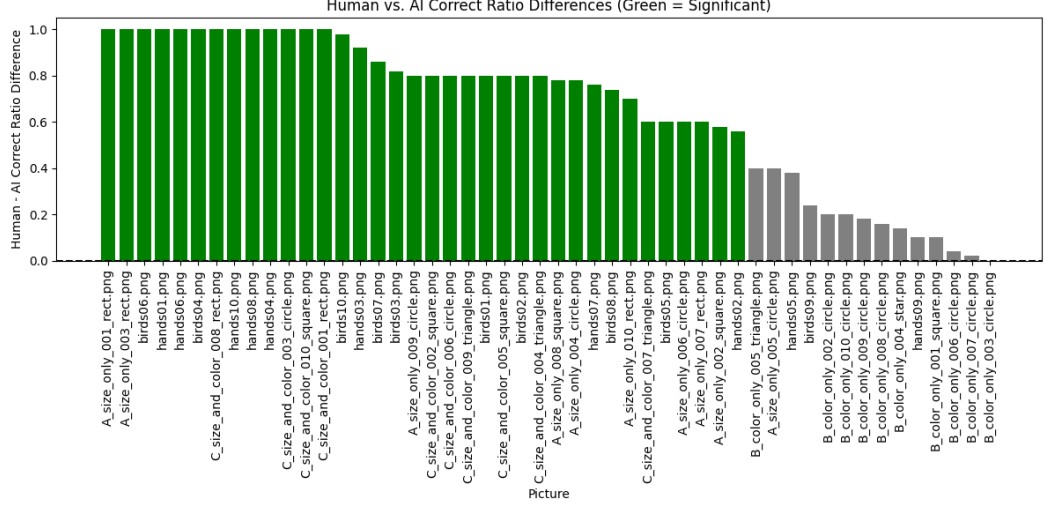

Figure 1: Human vs. AI correct-ratio differences (green bars are significant).

Table 1: Human vs. AI accuracy by task family.

| Task family | Human | AI | Gap (points) |
|---|---|---|---|
| Odd-one-out (size) | 78% | 6% | +72 |
| Odd-one-out (color) | 100% | 86% | +14 |
| Combined (size+color) | 86% | 0% | +86 |
| Bird-leg counting | 94% | 16% | +78 |
| Hand-finger counting | 98% | 21% | +77 |

## 4.5   Data Capture and Analysis

Data capture and analysis were scripted with LLM assistance (for API calls, answer parsing, and scoring), then manually verified by the authors. Statistical summaries included per-task accuracies, overall human–model gaps, and model-wise totals.

## 5   Results

**Overall performance.**   Figure 1 summarizes human minus AI accuracy per item (green markers: $p < 0.05$). Humans were near-ceiling on most items (task-family averages: A 78%, B 100%, C 86%, D 94%, E 98%). Models varied sharply by task type.

**(1) Color discrimination (B).** Models approached humans: five models averaged $\approx 85.6\%$ accuracy.

**(2) Size differences (A).** Performance collapsed: models averaged $\approx 5.6\%$ accuracy; humans 78%.

**(3) Combined distractors (C).** Models failed completely: across ten images, aggregate accuracy of all five models was 0%. Answers typically noticed the color outlier but ignored size. Humans achieved 86%.

**Commonsense-conflict tasks.** For bird-leg counting (D), models averaged $\approx 15.6\%$; for hand-finger counting (E), $\approx 20.8\%$. Models often defaulted to "two legs" or "five fingers" regardless of evidence. Humans were $\sim 94$–98%.

## 6   Discussion and Conclusions

The results support both H1 and H2. Models that excel at semantically cued color differences nonetheless fail on purely perceptual geometry (size) and on multi-feature integration (combined

Table 2: Overall accuracy across 50 items by model.

| Model (VLM) | Overall accuracy |
|---|---|
| GPT-5 | 33.4% |
| Claude-Sonnet-4-20250514 | 24.8% |
| Gemini 2.5-Pro | 31.4% |
| Doubao-seed-1.6-250615 | 20.4% |
| Qwen2.5-VL-72B-Instruct | 17.6% |

distractors), indicating an absent or weak low-level perceptual path. When visual evidence conflicts with prior knowledge, models strongly favor linguistic priors, echoing confirmation bias [Nickerson, 1998]. Accuracy is slightly higher on human hands than on birds (H1.1), plausibly reflecting training-data prevalence and targeted tuning for hands in modern generative pipelines. These findings align with evaluations reporting large human–model gaps on intuitive visual cognition [Schulze-Buschoff et al., 2024] and with visual-illusion CAPTCHAs that reliably trip models while remaining human-easy [Ding et al., 2025].

# 7    Implications for AI-Targeted CAPTCHAs

The task families here are simple for humans yet reliably difficult for today's models. A practical CAPTCHA could ask users to click the color outlier in a grid, identify the larger item, or count visible limbs in a photo. Curating a diverse bank of such items would create an effective human–AI separator in the near term. As models improve, the bank will require updates, but the present gap is substantive enough to be useful for security and for probing multimodal cognition.

# 8    Limitations and Future Work

Tasks and sample size are limited (five families, ten items each). Failures here do not imply universal deficits on all "intuitive" problems; prompt engineering, auxiliary perception modules, or multi-turn clarification might improve results. Future work should broaden task coverage (e.g., dynamic videos, 3D perception, cluttered scenes) and probe pathway usage more directly by manipulating prompts to emphasize visual evidence versus prior knowledge.

# 9    Ethical Statement

All images were AI-generated or drawn from public resources with no personal data and no animal experimentation. Five adult volunteers provided informed consent. LLM assistance was used for stimulus generation, data processing, and drafting under human supervision; final analyses and claims remain the authors' responsibility.

# 10    Reproducibility Statement

We provide details to enable independent replication: scripts for generating $5 \times 5$ grids (tasks A–C), prompt templates (Chinese), answer-parsing code, and scoring utilities, along with seeds and parameter ranges for color distances and size deltas. For tasks D–E, we release procedures and prompts that reproduce comparable AI-generated images while avoiding redistribution of any single long-lived CAPTCHA item. Each participant/model solved 50 items (10 per family); models were queried up to 10 attempts per item to assess stability. We report versions, prompts, decoding parameters, and significance tests (Fisher's exact, $\chi^2$). API-only inference; no training.

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

## A   Technical Appendices and Supplementary Material

Technical appendices with additional results, figures, graphs and proofs may be submitted with the paper submission before the full submission deadline, or as a separate PDF in the ZIP file before the supplementary material deadline. There is no page limit for the technical appendices.


